# Monoclonal Antibodies in Treating Chronic Spontaneous Urticaria: New Drugs for an Old Disease

**DOI:** 10.3390/jcm11154453

**Published:** 2022-07-30

**Authors:** Sara Manti, Alessandro Giallongo, Maria Papale, Giuseppe Fabio Parisi, Salvatore Leonardi

**Affiliations:** 1Pediatric Respiratory Unit, Department of Clinical and Experimental Medicine, San Marco Hospital, University of Catania, Via Santa Sofia 78, 95123 Catania, Italy; mariellapap@yahoo.it (M.P.); giuseppeparisi88@hotmail.it (G.F.P.); leonardi@unict.it (S.L.); 2Pediatric Unit, Department of Human and Pediatric Pathology “Gaetano Barresi”, AOUP G. Martino, University of Messina, Via Consolare Valeria, 1, 98124 Messina, Italy; 3Pediatric Unit, Maggiore Hospital, 97015 Modica, Italy; alegiallongo@hotmail.it

**Keywords:** monoclonal antibodies, biologics, chronic spontaneous urticaria, treatment, children, adults

## Abstract

**Background:** H1-antihistamines (H1AH) represent the current mainstay of treatment for chronic spontaneous urticaria (CSU). However, the response to H1AH is often unsatisfactory, even with increased doses. Therefore, guidelines recommend the use of omalizumab as an add-on treatment in refractory CSU. This paved the way for the investigation of targeted therapies, such as monoclonal antibodies (mAbs), in CSU. **Methods:** A literature review was conducted including papers published between 2009 and 2022 and ongoing trials about the efficacy and safety of mAbs as treatment for CSU. **Results:** Twenty-nine articles, a trial with preliminary results, and seventeen ongoing or completed clinical trials on the use of mAbs in CSU were included. Randomized controlled trials (RCTs), meta-analysis, and real-life studies have proven the effectiveness and safety of omalizumab as a third-line treatment in refractory CSU. However, a percentage of patients remain unresponsive to omalizumab. Therefore, other mAbs, targeting different pathways, have been used off-label in case series and others are under investigation in RCTs. Most of them have showed promising results. **Conclusions:** Omalizumab remains the best choice to treat refractory CSU. Although results from other mAbs seem to be encouraging to achieve symptom control in refractory CSU, thus improving patients’ QoL, RCTs are needed to confirm their effectiveness and safety.

## 1. Introduction

Urticaria is characterized by the development of wheals with or without angioedema. Chronic urticaria (CU) is defined as lasting for more than 6 weeks [1]. The prevalence of CU is estimated to be between 0.1 and 1.4% across different areas of the world [2,3].

Different triggers can elicit urticaria, such as cold, heat, contact, infections, and others. However, in 75% of the patients suffering from CU, the causal factor cannot be detected [4]. Accordingly, urticaria is defined as spontaneous when no specific trigger is identified [1].

Mast cells are primarily involved in the pathogenesis of chronic spontaneous urticaria (CSU) through the release of pro-inflammatory mediators, which, in turn, recruit neutrophils, eosinophils, and T lymphocytes [5,6]. Impaired intracellular signaling pathways, and type II and type I autoimmunity have been suggested as pathogenic mechanisms [7]. It has been found that 30–50% of patients with CSU produce immunoglobulin (Ig)G autoantibodies against IgE or its receptor (FcεRI), causing the degranulation of cutaneous mast cells and basophils, and thus histamine release [8,9]. Regarding type I autoimmunity, in a cluster of CSU patients, authors reported evidence of IgG and IgE against thyroperoxidase (TPO), defining this mechanism as “autoallergy”. Patients with CSU had a six-fold higher risk of TPO antibodies positivity than controls (odds ratio (OR) 6.72; 95% confidence interval (CI) 4.56, 9.89). However, their pathogenic role is still under debate [10,11,12,13]. More recently, the activation of cascade coagulation has been proposed as an alternative pathogenic mechanism, initiated by tissue factors expressed on eosinophils in lesional skin. This event leads to thrombin-mediated increased vascular permeability and mast-cell degranulation [14].

CU can significantly affect the health-related quality of life (HRQoL) of patients, as it has been reported to interfere with sleep quality, and school and work performance, especially in patients with uncontrolled disease, with subsequent high health care and indirect costs [15,16]. Notably, it has also been associated with psychiatric disorders, such as anxiety and depression [15].

The treatment of CU has been based on the avoidance or elimination of triggering factors and, when identified, on the treatment of the underlying causes, such as infection. The treatment of CSU is based on symptomatic drugs and, among these, second-generation H1-antihistamines (H1AH) represent the current mainstay of treatment according to guidelines [1]. Nevertheless, the percentage of non-responders to H1AH is around 60%, which remains high, despite the possibility of increasing the dose to four-fold the licensed dose (40–45% of non-responders to standard dose) [17,18,19]. Furthermore, the up-dosing of H1AH is not free from potential adverse effects in children [19]. Therefore, in the last two decades, new treatment approaches, including monoclonal antibodies (mAbs) and immunosuppressants (e.g., cyclosporine), have been introduced to optimize symptom control and improve HRQoL [20]. The identification of different molecular pathways underlying CSU has made them potential therapeutic targets [6,8]. In this context, mAbs represent targeted therapies directed towards specific molecular pathways, being potentially more efficacious and avoiding toxicity and/or side effects of immunosuppressants [21]. They have proven to be effective in other inflammatory and allergic diseases, such as rheumatoid arthritis and asthma [21,22,23]. In CSU, their use is currently restricted to moderate-to-severe forms refractory to standard treatment, and only omalizumab, an anti-IgE mAb, is labelled as an add-on treatment for CSU [24].

This review aims to assess the current literature on the efficacy and safety of omalizumab and other emerging biologics in treating CSU, both in the pediatric and adult populations.

## 2. Materials and Methods

### 2.1. Literature Review

We performed this literature review including papers published between January 2009 and February 2022. Two reviewers (S.M., A.G.) independently conducted searches of electronic medical literature databases, such as PubMed, Global Health, EMBASE. The search strategy is detailed in Search Strategy (Appendix A). Manual searches of the current literature were also performed by referring to Web of Science, Google Scholar, BMJ Best Practice, the World Health Organization (WHO), and Clinicaltrial.gov. The following variations and terms were used: “biologic drugs”, “biological”, “monoclonal antibody”, “treatment”, “omalizumab”, “anti-IgE”, “mepolizumab”, “anti-IL-5”, “dupilumab”, “tezepelumab”, “anti-thymic stromal lymphopoietin (TSLP)”, “rituximab”, “chronic spontaneous urticaria”, “child”, “children”, “adolescent”, and “adult”. Lastly, the selected references of included papers were searched to find any other relevant documents in accordance with the inclusion criteria.

### 2.2. Eligibility Criteria

The inclusion criteria were: any language, publication in peer reviewed journals, children and adults who have been diagnosed with CSU; original article, meta-analysis, systematic review, review, case series, case report, and letter about mAbs in treating CSU. Exclusion criteria were: original article, case series, case report, and letter not focusing on CSU treatment; guidelines.

### 2.3. Guideline Review

Two independent reviewers (S.M., A.G.) performed data extraction using standard templates to report recommendations in support of or against the use of biological drugs in treating CSU. Articles were excluded by title, abstract, or full text for irrelevance to the investigated issue.

## 3. Results

Twenty-nine articles and trials with preliminary results met the eligibility criteria (overall, 3110 patients) [25,26,27,28,29,30,31,32,33,34,35,36,37,38,39,40,41,42,43,44,45,46,47,48,49,50,51,52,53,54]. They will be discussed in the following paragraphs. Data are summarized in Table 1 and Table 2 [25,26,27,28,29,30,31,32,33,34,35,36,37,38,39,40,41,42,43,44,45,46,47,48,49,50,51,52,53,54]. Table 3 and Table 4 report studies focusing on outcomes related to disease severity and QoL, respectively [25,26,27,28,29,30,31,32,36,37,38,39,40,41,42,43,44,45,46,47,48,49,50,51,52,53,54]. Biologics and their target structures, receptors, and mediators tested for treating CSU are also represented in Figure 1.

We also included 17 ongoing or completed clinical trials on the use of mAbs in CSU (Table 5) [55,56,57,58,59,60,61,62,63,64,65,66,67,68,69,70,71].

### 3.1. Anti-IgE

#### 3.1.1. Omalizumab

Omalizumab is a humanized anti-IgE mAb that binds the constant region (cε3) of free IgE, preventing the interaction with high- and low-affinity IgE receptors (FcεRI and FcεRII). It reduces free IgE and downregulates FcεRI expression on basophils and mast cells, decreasing their degranulation and the subsequent release of mediators involved in the pathogenesis of CSU [72,73]. However, this mechanism of action does not explain fully all the effects of omalizumab in CSU, and other hypothesized mechanisms need to be further elucidated (e.g., effects on basopenia and coagulation abnormalities) [11].

Omalizumab was first approved to treat severe uncontrolled allergic asthma from age six and, subsequently, severe chronic rhinosinusitis (CRS) and CSU. At present, omalizumab is the only mAb licensed for the treatment of CSU in patients 12 years of age or older who remain symptomatic despite H1AH [24].

We included 10 randomized controlled trials (RCTs) (12 articles) testing omalizumab as an add-on treatment in a total of 2362 patients with moderate-to-severe CSU refractory to H1AH (Table 1) [25,26,27,28,29,30,31,32,33,34,35,36].

The first evidence of omalizumab efficacy in CSU was suggested in a case series of individuals treated for asthma, who reported improvement in CSU [74].

Successively, a phase II randomized double-blind placebo controlled trial (RDBPCT), MYSTIQUE, evaluated the administration of different doses of omalizumab in 90 adult patients. Changes from the baseline in weekly urticaria activity score (UAS7) appeared at week 1 and were already significant at week 4 in both the 300 mg and 600 mg groups compared with the placebo (−19.9 and −14.6 vs. −6.9 points; *p* < 0.001 and *p* = 0.047, respectively), while the 75 mg dose induced a non-significant change in UAS7 compared with the placebo. A plateau in dose–response was observed with around 300 mg omalizumab [25]. The authors suggested that the earlier onset of action in CSU than in asthma could be explained by lower total IgE levels and less-dependent IgE pathogenesis [25]. The change in UAS7 from baseline to week 24 was also significant in patients with moderate-to-severe CSU and positive IgE anti-TPO antibodies, which are probably involved in mast-cell degranulation, after omalizumab vs. placebo (−17.8 vs. −7.9 points; *p* = 0.0089). Two-thirds of patients in the treatment group reached the resolution of symptoms [26].

These trials paved the way for the development of further RCTs on larger populations. Among these, the results of three RDBPCTs, GLACIAL, ASTERIA I, and ASTERIA II, led to the approval of omalizumab for the treatment of CSU by the Food and Drug Administration (FDA) [27,28,29].

GLACIAL assessed the safety of omalizumab 300 mg as a primary endpoint, enrolling 335 individuals with CSU refractory to H1AH at up to four-fold the approved dose in combination with H2 antihistamines (H2AH) and/or leukotriene receptor antagonists (LTRAs). No difference in the rate of AEs was found between the treatment and placebo groups over 40 weeks (11% vs. 13%) [26]. The changes reported in weekly itch severity score (ISS7) at week 12 were significant (−8.6 vs. −4.0 points; *p* < 0.001), as similarly found for UAS7 and the dermatological quality of life index (DLQI) [27].

In ASTERIA I and II, in contrast with GLACIAL, the enrolled patients were symptomatic with H1AH at the approved dose, and other doses of omalizumab further than the 300 mg were tested [27,28,29].

ASTERIA II included 323 patients with moderate-to-severe refractory CSU and showed that 150 mg or 300 mg omalizumab significantly reduced ISS7 from baseline to week 12 compared with a placebo (−8.1 and −9.8 vs. −5.1 points; *p* = 0.001 and *p* < 0.001, respectively). The reduction was also significant for the secondary endpoints, such as UAS7, with 66% and 43% of patients in the 300 mg and 150 mg group, respectively, having a score of less than six [28].

The efficacy of omalizumab at week 12 in ASTERIA I (*n* = 319) was comparable with the above-mentioned trials with regard to ISS7, UAS7, and DLQI [27,28,29]. The reduction in ISS7 was statistically significant in the 300 mg and 150 mg groups compared with the placebo (−9.4 and −6.6 vs. −3.6 points; *p* < 0.0001 and *p* = 0.0012, respectively), as well as, in contrast with ASTERIA II, in the 75 mg group (−6.4 vs. −3.6 points; *p* = 0.001). The improvement of the analyzed outcome measures was sustained at week 24 [29].

The three above mentioned trials have shown a dose-dependent response for omalizumab, with higher rates of disease control or complete response (UAS ≤ 6 or UAS = 0, respectively), when patients were treated with 300 mg of omalizumab [27,28,29]. This dose was also associated with a higher percentage of sustained responses [27,28,74].

The POLARIS study was conducted on 218 Japanese and Korean individuals, who had a lower incidence of angioedema than the Caucasian population, and it confirmed the efficacy and safety of omalizumab 150 mg and 300 mg as an add-on treatment in refractory CSU, through the assessment of different outcome measures at week 12 (ISS7, *p* = 0.006 and *p* < 0.001, respectively; UAS7, *p* = 0.006 and *p* < 0.001, respectively; etc.). The effect of omalizumab was dose dependent [32]. A similar efficacy was observed in 418 Chinese adult patients (*p* < 0.001) [36].

The long-term efficacy of omalizumab has been less investigated thus far. A phase IV RDPCT, XTEND-CIU (Xolair Treatment Efficacy of Longer Duration in Chronic Idiopathic Urticaria), assessed this outcome. At first, all of the enrolled patients (*n* = 205) underwent 24 weeks of open-label treatment with omalizumab. Then, patients who achieved controlled disease (*n* = 134) were randomized into the omalizumab group or placebo group. A greater number of patients who were switched into the placebo group experienced a relapse of CSU compared with those who continued treatment (60% vs. 21%; *p* < 0.0001). In patients in whom omalizumab was re-started because of disease relapse, UAS7 was significantly reduced at week 12 (95% CI, −34.3 to −24.7; *p* < 0.0001). Interestingly, no significant difference was found in the incidence of relapse regardless of whether treatment was discontinued after 24 or 48 weeks (43.4% vs. 45.1% after 12 weeks discontinuation; *p* = 1) [33].

The OPTIMA trial, consistent with the XTEND-CIU trial, provided additional information about re-treatment with omalizumab in patients who had achieved controlled disease at week 24 (UAS7 ≤ 6). Among these, 48% relapsed after withdrawal and underwent 12 weeks of re-treatment, with 88% of them regaining disease control. It was also found that most of the patients (70%) with an inadequate response to treatment (UAS7 ≥ 6) (79% of the 150 mg group) achieved symptom control after increasing the dose to 300 mg. The same trend was observed in patients with inadequate disease control at week 24 (UAS7 ≥ 6), 22% of whom benefited of the extension treatment period, while the remaining generally had lower UAS7 scores than the baseline [35].


*QoL*


Changes in UAS7 and the chronic urticaria quality of life questionnaire (CU-Q2oL) or DLQI have been found to be closely correlated, meaning that changes in symptoms are reflected in an improved QoL [75].

As regards the effects of omalizumab on DLQI, the analysis of the three pivotal trials, ASTERIA I, ASTERIA II, and GLACIAL, showed a significant reduction from baseline at weeks 12 (ASTERIA II; *p* < 0.001) and 24 (ASTERIA I and GLACIAL; *p* < 0.05 and *p* < 0.0001, respectively) in the 300 mg group compared with the placebo, in patients both without and with angioedema. DLQI increased during follow-up, without reaching baseline values [27,28,29,76,77,78].

The XTEND-CIU trial reported similar findings. Treatment induced the improvement of DLQI, as well as other HRQoL measures, such as the insomnia severity index (ISI), and work productivity and activity Impairment (WPAI), at weeks 12 and 24, compared with baseline. The improvement was sustained during the following 24 weeks in the treatment group compared to the placebo group, who experienced a worsening in symptoms and quality of life (*p* < 0.0001) [34].

When the assessment of HRQoL outcomes was restricted to CSU patients with angioedema, which has a remarkable impact on a patient’s quality of life, significant improvement was still reported in the CU-Q2oL score, DLQI, and the angioedema quality of life questionnaire (AE-QoL) at weeks 4 and 28 compared with the placebo (*p* < 0.001) [30,31].


*Safety*


GLACIAL, whom primary endpoint was to assess the safety of omalizumab in refractory CSU, showed that the rates of adverse events (AEs) and serious adverse events (SAEs) over the 24 weeks of treatment and 16 weeks of follow-up were similar compared with the placebo. Moreover, no episodes of anaphylaxis were reported [27]. In ASTERIA II, a greater incidence of SAEs was observed in the 300 mg group (6% vs. 3%) [28]. However, these were reported mainly during the follow-up when treatment had already been discontinued [28].

Consistent with GLACIAL, no safety concern emerged from the other trials, even in the event of a longer treatment duration as in the XTEND-CIU trial. The most common reported AEs were headaches and nasopharyngitis [33].

#### 3.1.2. Ligelizumab

Ligelizumab is an anti-IgE mAb binding IgE, preventing their bound to high-affinity IgE receptors (FcεRIα) on mast cells and basophils, needed for the release of inflammatory mediators, with an in vivo nine-fold higher affinity to IgE (95% CI 6.1–14) compared with omalizumab [79]. Hence, it is expected that it should be more efficacious in Fc-RI-driven diseases such as CSU than omalizumab, which induces a greater inhibition of IgE-binding to CD23 involved in antigen presentation. Furthermore, the suppression of the skin prick response to allergens in atopic individuals was also significantly higher after ligelizumab than omalizumab (*p* < 0.001) [79,80].

Ligelizumab, differently from omalizumab, is currently not labelled for any disease, though several trials are ongoing (NCT03907878, NCT04210843, NCT04513548, NCT03580369, NCT03580356) or have been performed to evaluate its effectiveness and/or safety in CSU (Table 2 and Table 3) [37,38,39,40,55,56,57,58,59].

We included two RDBPCTs, the extension phases of one of these trials, and a trial with preliminary results available [37,38,39,40]. The largest RDBPCT recruited 382 adults with moderate-to-severe CSU poorly controlled by H1AH alone or in combination with H2AH or LTRAs. Patients were randomized into six treatment groups (240 mg, 72 mg, 24 mg monthly ligelizumab, 300 mg monthly omalizumab, one ligelizumab 120 mg dose followed by placebo, and placebo). At week 12, the percentage of patients with a complete response to treatment (weekly hives severity score (HSS7) = 0) was higher in the ligelizumab subgroups compared with the omalizumab group (51% and 42% in the 72 mg and 240 mg ligelizumab groups, respectively, vs. 26% in the omalizumab group). Similar rates were reported as regards UAS7 = 0. Furthermore, patients treated with 240 mg ligelizumab showed a longer-lasting response after treatment discontinuation [37]. The assessed HRQoL outcomes, such as DLQI and sleep and work interference, were all improved compared with the baseline. The extension phase of this study included 226 patients with a UAS7 ≥ 12 at week 32. They underwent 52 weeks of treatment with ligelizumab 240 mg, experiencing sustained improvement of the abovementioned outcomes [38]. Notably, the extension phase of this trial showed the long-term safety and efficacy of ligelizumab. Indeed, half of the patients—who had showed a poor response in the first phase of the study—reached a complete response at week 12 and 52 (46.5% and 53%, respectively) and disease control was confirmed to be long-lasting even after treatment suspension, with a median time to relapse of 38 weeks [39].

The other RDBPCT (NCT03437278) was conducted on 49 adolescents (12–17 years old) with treatment-refractory CSU to investigate ligelizumab (24 mg, 120 mg, 8-weeks placebo followed by 120 mg ligelizumab), as an add-on treatment to H1AH for 24 weeks. The preliminary results showed that all three groups reported a reduction from the baseline in UAS7, ISS7, HSS7, and DLQI at different endpoints (week 12, 24, and 40) [40].

As regards the safety profile, no serious adverse events were reported in around 900 individuals treated with ligelizumab [81].

#### 3.1.3. Quilizumab

Quilizumab is an afucosylated anti-IgE mAb directed against the M1 segment of IgE, which is expressed only on surface IgE, with a higher affinity than the fucosylated type. Quilizumab was able to determine IgE-switched B cells apoptosis in vitro, thus reducing IgE production [82]. In patients with asthma and allergic rhinitis, the reduction in IgE levels was significant and long lasting [83].

In an RDBPCT on 32 patients with CSU refractory to H1AH, quilizumab induced a reduction in the serum total IgE. Nevertheless, no significant change was observed in clinical scores at week 20 compared with the placebo (ISS7 = −3.3 points (90% CI, −7.2 to 0.7; *p* = 0.17) and UAS7 = −5.8 points (90% CI, −14.1 to 2.5; *p* = 0.24)). The authors suggested that this may be due to the different mechanism of action of quilizumab compared with omalizumab, or an inappropriate/inadequate dosage of quilizumab (Table 2) [41]. No other trial on quilizumab is ongoing.

#### 3.1.4. Other Anti-IgE

GI−301 and UB−221 are long-acting anti-IgE mAbs [84]. An open-label dose-escalating trial (NCT03632291) to assess the safety of UB-221 in CSU has been completed, and two other trials (NCT05298215, NCT04175704) will investigate UB-221 as an add-on treatment in CSU [60,85,86].

### 3.2. Anti-IL-5

The IL-5 signaling pathway is involved in B-cells’ and eosinophils’ proliferation, maturation, and survival [87,88].

Since eosinophilic inflammation represents a specific endotype in asthma and other Th2-driven diseases [89], mAbs targeting IL-5 (mepolizumab and reslizumab) or IL-5R (benralizumab) have been successfully tested to treat severe refractory eosinophilic asthma, representing a therapeutic option [90].

In the inflammatory response of CSU, which also shows a Th2 inflammation pattern, mast cells, together with B cells and basophils, play a central role, by releasing different mediators, one of which is IL-5 [6,91]. Eosinophils, in turn, can be responsible for mast-cell degranulation in CSU, as well as tissue destruction mediated by the major basic protein [92]. Moreover, the evidence of eosinophilic infiltration in the lesional and non-lesional skin of patients with CSU, without blood eosinophilia, suggested their pathogenic role in such diseases [93]. Interestingly, around 10% of patients with CSU may present blood eosinopenia (<0.05 × 10^9^/L) that has been associated with more severe disease [92].

On this basis, the use of anti-IL-5 mAbs has been suggested in CSU. We included four studies: a clinical trial on bernalizumab, a case report on mepolizumab, and a case report on reslizumab (Table 2) [42,43,44]. Overall, 14 patients underwent treatment with an anti-IL-5 agent [42,43,44]. Notably, patients treated with mepolizumab and reslizumab suffered from severe refractory eosinophilic asthma and comorbid refractory CSU [42,43].

#### 3.2.1. Mepolizumab

Mepolizumab is currently approved by the European Medicine Agency (EMA) to treat severe refractory eosinophilic asthma (≥6 years old), severe CRS with nasal polyps, uncontrolled eosinophilic granulomatosis with polyangiitis, and hypereosinophilic syndrome [94]. Interestingly, mepolizumab, when used to treat patients with severe eosinophilic asthma and concomitant CU (induced by NSAIDs), resulted in a rapid remission in urticaria and a significant reduction in eosinophil blood count [95].

In a case report, a German woman affected by severe eosinophilic asthma and concomitant refractory CSU had a complete response after treatment with mepolizumab for 16 weeks. However, when she interrupted treatment because of an immune complex reaction, urticaria symptoms relapsed [42]. A single-arm open-label trial is investigating the efficacy of mepolizumab in refractory CSU (NCT03494881) [61].

#### 3.2.2. Reslizumab

Reslizumab is approved for adults with severe asthma that is not properly controlled by a combination of inhaled high-dose corticosteroids plus another medicine used for the prevention of asthma [96]. In a 43-year-old patient with severe refractory eosinophilic asthma and refractory CSU and cold urticaria, reslizumab was shown to induce a sustained improvement of symptoms during 5 months of treatment [43].

#### 3.2.3. Benralizumab

Benralizumab is an anti-IL-5 receptor (IL-5R) mAb indicated as an add-on maintenance treatment of severe uncontrolled eosinophilic asthma in patients 12 years old and older [97]. Information from only one trial is available regarding benralizumab in CSU. It involved 12 adult patients with moderate-to-severe CSU inadequately controlled by H1AH. They received monthly benralizumab for 3 months, after a placebo dose. Nine patients completed the treatment, showing a significant reduction from baseline in UAS7 score at week 20 (95% CI, −6.6 to −24.8; *p* < 0.001), with no AEs reported. Five of them had a complete response (UAS7 = 0) at week 24 [44]. A phase II RDBPCT (NCT04612725) on benralizumab is still ongoing in patients with H1AH-refractory CSU [62].

### 3.3. Anti-IL-4 Receptor

#### Dupilumab

Dupilumab is a mAb that targets the IL-4-receptor alpha chain (IL-4Rα), antagonizing IL-4 and IL-13, two cytokines involved in the Th2 inflammatory pathway. Notably, IL-4 induces Th2 cell differentiation and B cell class-switching to IgE [6,98].

Although the underlying pathogenic mechanism of CSU is mainly characterized by Th2-driven responses, dupilumab is currently approved by the EMA for individuals with moderate-to-severe uncontrolled atopic dermatitis (≥12 years old or, if severe, ≥6 years old), severe uncontrolled asthma (≥6 years old), and CRS with nasal polyposis [99]. Recently, the FDA also approved dupilumab for the treatment of moderate-to-severe atopic dermatitis from the age of 6 months and eosinophilic esophagitis in patients aged 12 years or older and weighing at least 40 kg [100]. The use of dupilumab is still off-label in CSU and data are limited to three case series, including a total of 10 patients, suggesting dupilumab may be an effective treatment for refractory CSU (Table 2) [45,46,47].

Six patients with CSU refractory to concomitant treatment with antihistamines and omalizumab (300 mg and/or 600 mg) and comorbid atopic dermatitis underwent treatment with dupilumab in combination with other drugs (H1AH and/or topical tacrolimus and/or montelukast). Five patients out of six reached controlled disease with a UAS7 ≤ 6, when reported, or the resolution of CSU at month three [45]. Staubach et al. reported two children, aged 6 and 17 years old, respectively, suffering from CSU and treated with dupilumab after a poor response to up-dosing omalizumab and cyclosporine. Both patients had improved symptoms, especially the one with high IgE levels [46].

A similar effectiveness of dupilumab was observed in two adult women with refractory CSU, one of them was treated with dupilumab for comorbid atopic dermatitis, who had a complete and sustained response to treatment [47].

At present, two RDBPCTs, DUPISCU and CUPID (NCT03749135 and NCT04180488, respectively), are investigating the efficacy of dupilumab in 456 individuals with moderate-to-severe CSU refractory to H1AH [63,64].

### 3.4. Anti-CD20

#### Rituximab

Rituximab is a chimeric murine–human recombinant mAb that binds to CD20, which is expressed on the cell surface of B lymphocytes. Its mechanism of action consists of the depletion of B cells [101]. Rituximab-opsonized B cells have been suggested as an additional mechanism, reducing the interaction of macrophages with immune-complexes [102].

Rituximab is currently licensed to treat malignancies (non-Hodgkin’s lymphoma and chronic lymphocytic leukemia), autoimmune disorders (rheumatoid arthritis), vasculitis (granulomatosis with polyangiitis, microscopic polyangiitis), and dermatological conditions (pemphigus vulgaris) [103].

However, it has been increasingly used off-label in other autoimmune diseases (e.g., multiple sclerosis) [104].

Authors have reported the efficacy of rituximab 375 mg/m^2^ or 1000 mg in treating patients affected by CSU refractory to H1AH variably combined with other drugs (corticosteroids, immunosuppressants, and omalizumab) [48,49,50,51]. All patients (*n* = 4) achieved a complete remission early on and maintained it for several months after receiving rituximab [48,49,50,51]. The long-lasting remission induced by rituximab may be explained by B-cell depletion with a subsequent reduction in the synthesis of IgG autoantibodies, which have been suggested to be involved in the pathogenesis of CSU [105].

However, the reduction in Ig levels caused by rituximab can be associated with an increased risk of infections [106].

An open-label trial (NCT00216762) designed to assess the efficacy and safety of rituximab in 15 individuals with refractory CSU was stopped early and data are not available [65].

### 3.5. Anti-IL-17

#### Secukinumab

In patients with CSU, IL-17 expression has been found to be significantly higher in CD4+ lymphocytes and mast cells in both lesional and non-lesional skin biopsies compared with healthy controls (*p* < 0.0001) [52]. Patients with CSU have also shown blood IL-17 concentrations higher than controls. Interestingly, IL-17 levels were higher in patients with severe disease [107]. Secukinumab, neutralizing IL-17, is currently labelled to treat moderate-to-severe psoriasis and spondilarthritis [108]. It was found to be strongly effective in eight patients affected by severe CSU refractory to omalizumab. The patients had a baseline UAS7 ranging from 32 to 40, much higher than patients recruited in other studies, and after three months of treatment reported an 82% reduction in UAS7 [52].

### 3.6. Anti-IL-1

#### Canakinumab

Canakinumab antagonizes IL-1β, an essential cytokine to innate immunity [109]. In an RDPCT (NCT01635127), one single dose of canakinumab 150 mg did not induce significant changes in UAS7 at week 4 in 20 adults with refractory CSU. Hence, the authors concluded that IL-1β might not be involved in the pathogenesis of CSU [53].

### 3.7. Anti-Tumor Necrosis Factore Alfa (TNF-α)

#### Infliximab

TNF-α has been reported to increase in the lesional and non-lesional skin of patients with cold and pressure urticaria, as well as in the serum of patients with CU, correlating with disease severity [110,111]. A case report described the use of the TNF-α inhibitor infliximab in a 35-year-old woman suffering from CSU. The patient experienced symptom resolution for 5 months, until treatment was switched to etanercept. Re-treatment with infliximab was effective again for a further three years, when flares were controlled by adding cyclosporine [54].

### 3.8. Anti-TSLP

#### Tezepelumab

TSLP, IL-25, and IL-33 are released following different triggers on epithelia. They start the Th2 inflammatory response, mediating T-cell polarization in Th2 cells [112,113,114].

Tezepelumab, a human monoclonal antibody inhibiting the action of TSLP, appears to prevent and treat the lesional skin of patients with CSU [6]. Its efficacy and safety are currently tested in a phase II, multi-center, interventional, randomized, parallel-group, placebo-controlled, omalizumab-controlled clinical trial enrolling 175 adult participants (18 to 80 years) with CSU [66].

### 3.9. Other Biologics

#### 3.9.1. Barzolvolimab

Barzolvolimab (CDX-0159) antagonizes the tyrosine kinase receptor KIT, whose ligand is the stem cell factor. The KIT receptor pathway is involved in mast-cell differentiation. One dose of Barzolvolimab was able to suppress mast cells in healthy individuals and two RDBCTs (NCT04538794, NCT05368285) are recruiting patients with refractory CSU to assess its efficacy and safety [67,68,115].

#### 3.9.2. MTPS9579A

MTPS9579A is anti-tryptase mAb that acts by dissociating tetramers into inactive monomers [116]. Tryptase is the main mediator accumulated in mast-cell granules and, when released, promotes and amplifies the inflammatory response [117]. The administration of MTPS9579A resulted in a dose-dependent reduction in active tryptase in the upper airways of 106 healthy individuals [116]. An RDBPCT (NCT05129423) is enrolling patients to evaluate MTPS9579A as a treatment for refractory CSU [69].

#### 3.9.3. LY3454738

LY3454738 is an agonist of the CD200 receptor that is associated with Th2 inflammation. Indeed, CD200R expression is higher on Th2 cells, and it is upregulated by IL-4. Its expression is much greater in peanut-specific CD^+^ T cells of patients with an allergy to peanuts [118]. LY3454738—due to its suggested role in Th2 inflammation—was investigated in an RDBPCT on patients with refractory CSU (NCT04159701) that was stopped early because of a lack of efficacy [70].

#### 3.9.4. Lirentelimab

Lirentelimab (AK002) acts as an anti-sialic acid-binding, immunoglobulin-like lectin (Siglec)-8, an inhibitory receptor of the CD33-related family selectively expressed on eosinophils and mast cells. AK002 has been shown to inhibit mast cells and to induce the apoptosis of eosinophils [119]. An open-label study (NCT03436797) has been conducted to determine the efficacy and safety of AK002 in refractory CU. However, the results are not yet available [71].

## 4. Discussion

This review provides an overview of the currently available evidence regarding the use of mAbs as a treatment for CSU (Table 1, Table 2, Table 3, Table 4 and Table 5) [25,26,27,28,29,30,31,32,33,34,35,36,37,38,39,40,41,42,43,44,45,46,47,48,49,50,51,52,53,54,55,56,57,58,59,60,61,62,63,64,65,66,67,68,69,70,71].

Although omalizumab still remains the only approved mAb in treating CSU, other biologics have shown promising results and are currently under investigation in several trials [24,37,38,39,40,42,43,44,45,46,47,48,49,50,51,52,54,55,56,57,58,59,60,61,62,63,64,65,66,67,68,69,70,71].

Regarding omalizumab, a number of performed trials with a consistent number of enrolled patients have shown that omalizumab is effective, improving disease control and QoL, and safe, thus representing a well-established add-on treatment in refractory CSU, as stated by the updated EAACI guidelines [1,25,26,27,28,29,30,31,32,33,34,35,36]. Nevertheless, a limitation is the lack of RCTs on children. The only data refer to adolescents (≥12 years of age), who have been included in RCTs with adults, where they represent a marginal percentage, and they are not analyzed separately. A prospective open-label study on 29 adolescents with refractory CSU confirmed the effectiveness of omalizumab, with 58% of patients reaching a complete response (UAS = 0) at week 12. Three patients had a relapse after several months (from 4 to 12) following omalizumab withdrawal [120]. A review including 13 children reported a complete response in 12 of them after omalizumab 150 mg or 300 mg [121].

An analysis of 67 prospective observational studies, systematic reviews, and meta-analysis on RDPCTs added further data about the effectiveness and safety of omalizumab in adolescents and adults with CSU [122,123,124].

Nevertheless, around 30–40% of patients do not achieve disease control (UAS ≤ 6) with omalizumab [26,27,125]. This might be due to the standard dose of omalizumab, not adapted to weight and IgE levels, as seen in asthma, and/or high IgE levels (>1500 IU/mL), and/or different pathogenic mechanisms [45,46,47,48,49,50,51,52,53,54,55,56,57,58,59,60,61,62,63,64,65,66,67,68,69,70,71,72,73,74,75,76,77,78,79,80,81,82,83,84,85,86,87,88,89,90,91,92,93,94,95,96,97,98,99,100,101,102,103,104,105,106,107,108,109,110,111,112,113,114,115,116,117,118,119,120,121,122,123,124,125,126]. Omalizumab up-dosing to 600 mg reduced the proportion of non-responders to 7% [127].

With the aim of optimizing a treatment, high total serum IgE levels have been suggested as a biomarker predictive of the response to omalizumab [128]. Indeed, patients who exhibited a poor response to omalizumab had lower pre-treatment IgE levels compared with responders, who also showed an increase in IgE levels at week 4, and the IgE level at week 4/IgE level at baseline ratio revealed its superiority as a predictor of the response to treatment [129]. Blood basophils and histamine, which both increased in patients treated with omalizumab 300 mg, could represent other biomarkers predictive of the response to treatment [130]. Serum transglutaminase-2 activity may be a more reliable monitoring biomarker of the response to omalizumab, being less influenced by other comorbidities than IgE [131].

Another unanswered question concerns the optimal duration of treatment. RCTs have reported CSU-relapses after the interruption of omalizumab, with a subsequent response when treatment was re-started [33]. Therefore, omalizumab cannot be defined as a disease-modifying drug, and long-term treatments seem to be needed to control the disease.

Ligelizumab (240 mg), another anti-IgE, drug has shown superiority to omalizumab, probably due to its slightly different mechanism of action and higher affinity to IgE [37]. However, at the end of 2021, Novartis announced that ligelizumab showed superiority to a placebo, but not versus omalizumab at week 12 in two ongoing trials (NCT03580369 and NCT03580356), although the data are not yet available [58,59,132]. Contrary to this, quilizumab did not improve symptoms [41].

Currently, several experimental and clinical research studies are ongoing with the aim to provide further evidence on the pathogenesis of CSU. Understanding the close relationship between pathogenic pathways and clinical features will allow the identification of novel predictive biomarkers helpful in selecting the best candidate to receive targeted therapies with mAbs, and, consequently, the achievement of better clinical outcomes.

In addition to IgE, other investigated targets have included IL-5/IL-5R, through the development of anti-IL-5 mAbs (mepolizumab, reslizumab, and benralizumab), showing efficacy in 14 patients [42,43,44,61,62]. The IL-4 and IL-17 pathways, targeted by dupilumab and secukinumab, respectively, seem to play a remarkable role in the pathogenesis of CSU; thus, they could be an additional therapeutic weapon in the treatment of refractory CSU [6,52,107]. Nevertheless, data on these mAbs, though encouraging, come from case series, thus no firm conclusions can be drawn about their efficacy [45,46,47,52]. Ongoing and future RCTs on larger populations will clarify their potential therapeutic role in CSU.

TSLP, IL-25, and IL-33, the so called “alarmins” probably represent one of the most intriguing targets because they are located upstream of the inflammatory cascade. Hence, blocking the alarmins pathway could potentially be more efficacious and modify the disease course [114]. Barzolvolimab, suppressing mast cells, could represent another disease-modifying drug [115]. Although it is not the purpose of this review, it is necessary to mention that, among biologic drugs, small molecule inhibitors such as remibrutinib (LOU064), a Bruton’s tyrosine kinase (BTK) inhibitor with a potential role in the treatment of CSU, represent an alternative to mAbs [133]. Remibrutinib, similar to other BTK inhibitors (fenebrutinib, tirabrutinib, rilzabrutinib, and TAS5315), targets BTK, which is involved in B-cell differentiation and proliferation and mast-cell activation, mediated by B-cell receptor and FcεRI activation, respectively (Table 6 and Table 7) [133,134,135,136,137,138,139,140,141,142,143,144]. Remibrutinib at different doses showed superiority to a placebo in the NCT03926611 trial [133]. Similarly, the preliminary results of the NCT03137069 trial on fenebrutinib 150 mg daily and 200 mg twice a day showed a significant reduction from the baseline in UAS7 at week 8 compared with a placebo (−17.6 and −20.7, respectively, vs. −11.2) [145]. On the contrary, trials on tirabrutinib and etanercept, a TNF-α antagonist, have been stopped early [141,144]. Other trials are ongoing to investigate inhibitors acting on different targets, such as JAK1/TYK2 and prostaglandin D2 receptor 2 (DP2 or CRTH2) (Table 6 and Table 7) [146,147]. CRTH2 plays a role in the chemotaxis of Th2 cells and eosinophils, and Th2 cytokine synthesis. AZD1981, a CRTH2 antagonist, induced a significant reduction in UAS7 at the end of the drug wash out period compared with a placebo and with no safety concern [147]. To summarize, small molecule inhibitors may represent an alternative to mAbs as targeted therapies in refractory CSU, with the advantage for some of them of oral administration compared with mAbs. However, data on inhibitors, excepted for etanercept, whose use has been reported successfully in a case report of CSU, are limited to few trials, that, to date, do not allow us to draw conclusions on their efficacy and safety [54,135,141,147].

## 5. Conclusions

In line with the current guidelines [1], omalizumab has been demonstrated as an effective and safe treatment, allowing a remarkable advance in the management of CSU. Our center’s experience is consistent with data on the efficacy of omalizumab, although it is limited to the treatment of relatively few patients. Other biological drugs have shown promising results in treating CSU, and those acting upstream of the inflammatory cascade, such as dupilumab and tezepelumab, may be of major interest and efficacy in the future. “New” mAbs may allow the creation of individualized targeted and more efficacious therapies in patients with treatment-refractory CSU to achieve symptom control, thus improving patients’ QoL. In this context, H1AH may maintain a role, mainly as a rescue medication in the event of relapses. However, further RCTs on a larger scale are needed to identify biomarkers able to predict the response to treatment, the optimal dosage, and the duration of treatment for each mAb, and to assess their long-term effectiveness and safety, both in children and adults, as well as the most appropriate management of the CSU patient after the withdrawal of biological drugs.

## Figures and Tables

**Figure 1 jcm-11-04453-f001:**
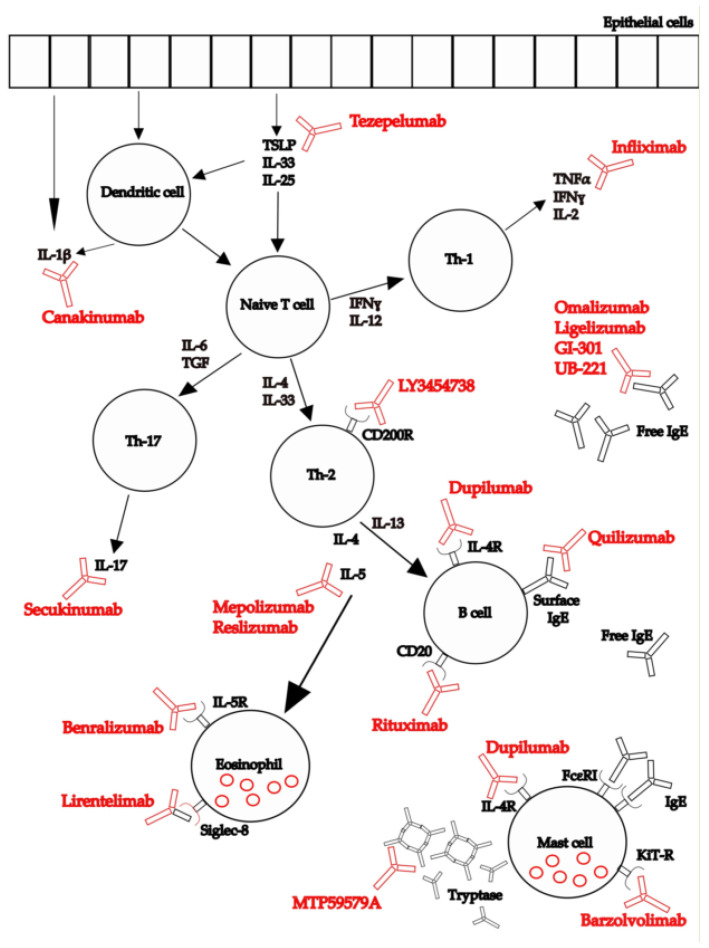
**Biologics for chronic spontaneous urticaria (CSU).** Biologics and their target structures, receptors and mediators tested for treating CSU.

**Table 1 jcm-11-04453-t001:** List of the studies investigating omalizumab in treating CSU.

Authors	Type of Study	N.	Age (Yrs)	Indication	Dosage	Duration	Follow-Up	Results	Adverse Events	Beneficial
Saini et al. 2011 [25]	Phase 2 RDBPCT MYSTIQUE	90	40.8 ± 14.7 <18 (5.6%)	Moderate-to-severe CSU despite H1AH UAS7 ≥ 12	75, 300, 600 mg or placebo combined with H1AH as needed	4 weeks	12 weeks	At week 4 ↓ ΔUAS7 −6.9 (placebo), −9.8 (75 mg) −19.9 (300 mg), −14.6 (600 mg)	44% ≥ 1 AE ≅AEs vs. placebo URTI, Pharyngitis Headache 2 hypersensitivity	Yes
Maurer et al. 2011 [26]	RDBPCT X-QUISITE	49	40.5 (18–70)	CSU refractory to H1AH UAS7 ≥ 10 Total IgE 30–700 IU/mL anti-TPO IgE ≥ 5.0 IU/mL	75–375 mg Q2W or Q4W or placebo	24 weeks	Not reported	At week 24 ↓ UAS7 (−17.8 vs. −7.9) symptoms free (67 vs. 4%) ↓ concomitant medication use	≅AEs vs. placebo Diarrhea Pharyngitis Headache	Yes
Kaplan et al. 2013 [27]	Phase 3 RDBPCT GLACIAL	335	43 ± 14	CSU refractory to H1AH (up to x4) + H2AH or LTRAs, or both UAS7 ≥ 16	300 mg or placebo Q4W as add-on	24 weeks	16 weeks	No safety concern (w 40 w) At week 12 ↓ ISS7 (−8.6 vs. −4) ↓ UAS7 (−19 vs. −8.5)	≅ incidence of drug related AEs vs. placebo (11 vs. 13%)	Yes
Maurer et al. 2013 [28]	RDBPCT ASTERIA II	323	42.5 ± 13.7 (≥12)	Moderate-to-severe CSU symptomatic despite H1AH UAS7 ≥ 16	75 mg, 150 mg, 300 mg or placebo Q4W + H1AH	12 weeks	16 weeks	At week 12 ↓ ISS7 (−5.9 vs. −8.1 vs. −9.8) ↓ UAS7 ↑ QoL	≅rate of AEs Higher rate of SAEs (6%) in 300 mg group	Yes
Saini et al. 2015 [29]	Phase 3 RDBPCT ASTERIA I	319	41 (12–75) 12–17 (2.5%)	CSU refractory to H1AH UAS7 ≥ 16	75 mg, 150 mg, or 300 mg or placebo Q4W	24 weeks	16 weeks	At week 12 ↓ ISS7 −9.4 (300 mg), −6.6 (150 mg) −6.4 (75 mg), −3.6 (placebo) ↓ UAS7 ↓ rescue medicine	Mild dose-dependent AEs Headache Arthralgia Injection-site reactions	Yes
Staubach et al. 2016 [30]	Phase 3 RDBPCT X-ACT	91	42 ± 12 (18–75)	CSU with angioedema refractory to 2–4x H1AH UAS7 ≥ 14 CU-Q2oL score ≥ 30	300 mg or placebo Q4W	28 weeks	8 weeks	At week 28 ↓ AE-QoL and AAS7 At week 12 ↓ UAS7 (−16 vs. −4)	≅AEs vs. placebo SEAs not drug-related	Yes
Staubach et al. 2017 [31]	RDBPCT X-ACT	91	42 ± 12 (18–75)	CSU with angioedema (≥4 episodes in 6 months) refractory to 2–4x H1AH UAS7 ≥ 14	300 mg or placebo Q4W	28 weeks	8 weeks	↓ AE-QoL ↓ DLQI ↓ AAS7	NR	Yes
Hide et al. 2017 [32]	Phase 3 RDBPCT POLARIS	218	43.5	CSU refractory to standard H1AH UAS7 ≥ 16	300 mg, 150 mg, or placebo Q4W With H1AH	12 weeks	12 weeks	At week 12 ↓ ISS 7 −10.2 (300 mg), −8.8 (150 mg) −6.5 (placebo) ↓ UAS7	≅AEs vs. placebo Pharyngitis Headache, Eczema 1 pharyngeal edema	Yes
Maurer et al. 2017 [33]	RDBPCT XTEND-CIU	205 (open label) 134 (double blind)	44 ± 14 (open label) 45 (double blind) (12–75)	CSU refractory to H1AH UAS7 ≥ 16	300 mg Q4W for 24 weeks then randomization if UAS7 ≤ 6 If UAS7 ≥ 12 at week 24–48 -> Re-trt with omalizumab	48 weeks	12 weeks	↑ UAS7 and DLQI after discontinuation or placebo UAS7 ≥ 12 week 24–48 (21% omalizumab vs. 60% placebo) ↓ UAS7 after re-treatment	16 drug-related AEs 6 SAEs not drug-related 1 anaphylaxis	Yes
Casale et al. 2019 [34]	Open-label + RDBPCT XTEND-CIU	205 (open label) 134 (double blind)	44 ± 14 (open label) 45 (double blind) (12–75)	CSU refractory to H1AH 48% CSS	300 mg Q4W for 24 weeks then randomization if UAS7 ≤ 6 If UAS7 ≥ 12 at week 24–48 -> open label omalizumab	48 weeks	12 weeks	At week 12 and 24 ↓ HRQoL scores At week 48 Sustained improvement of HRQoL scores	NR	Yes
Sussman et al. 2020 [35]	Phase 3 RCT OPTIMA trial	314	46.3	CSU refractory to H1AH, H2AH, LTRA	150 mg or 300 mg Q4W Step-up to 300 mg if UAS7 ≥ 6 before week 24	24 weeks (+12 weeks if UAS7 ≥ 6) 12 weeks re-trt if UAS ≥ 16	4–24 weeks	At week 24 Step-up to 300 mg (79% in 150 mg) UAS7 ≥ 6 (31% in 300 mg) UAS ≤ 6 (37%) -> UAS7 ≥ 6 after discontinuation (48%) -> re-trt -> UAS7 ≤ 6 (88%)	13% ≥ 1 AEs Headache, Pharyngitis Nausea Fatigue 8 SAEs not drug-related	Yes
Yuan et al. 2022 [36]	RDBPCT	418	≥18	CSU refractory to H1AH for ≥6 months	150 or 300 or placebo mg Q4W	20 weeks	NR	At week 12 ↓ ISS 7 (LSM) −4.2 (300 mg), −3.8 (150 mg) −2.3 (placebo)	A little higher AEs in 300 mg (71 vs. 64%)	Yes

AAS7, weekly angioedema activity score; CSS, corticosteroids; Q4W, every 4 weeks; LSM, least square means; N., number of patients; NR, not reported; trt, treatment; URTI, upper respiratory tract infection.

**Table 2 jcm-11-04453-t002:** List of the studies investigating other mAbs in treating CSU.

Ligelizumab										
Authors	Type of Study	N.	Age (Yrs)	Indication	Dosage	Duration	Follow-Up	Results	Adverse Events	Beneficial
Maurer et al. 2019 [37]	Phase 2b RDBPCT NCT02477332	382	43.3 ± 12.5 (18–75)	CSU refractory to H1AH ± H2AH ± LTRA UAS7 ≥ 16 HSS7 ≥ 8	Ligelizumab 240 or 72 or 24 mg Q4W, or omalizumab 300 mg Q4W; placebo Q4W; 120 mg ligelizumab followed by placebo Q4W Combined with standard trt	20 weeks	24 weeks	dose–response curve plateau at 72 mg dose ligelizumab HSS7 = 0 at week 12 72 mg ligeliz > omaliz (51 vs. 26%) 240 mg ligeliz > omaliz (42 vs. 26%) UAS7 = 0 at week 12 72 mg ligeliz > omaliz (44 vs. 26%) 240 mg ligeliz > omaliz (40 vs. 26%) at week 20 240 mg ligeliz > omalizumab	≅incidence of AEs ↑ injection site reactions in 72 mg and 240 mg ligelizumab groups Headache	Yes
Giménez-Arnau et al. 2022 [38]	Open-label extension study of NCT02477332	226	44.5 ± 12.7 (≥18)	UAS7 ≥ 12 at week 32 in NCT02477332	240 mg Q4W	52 weeks	48 weeks	↓ SIS7 ↓ AIS7 ↓ work impairment	NR	Yes
Maurer M et al. 2021 [39]	Open-label extension study of NCT02477332	226	44.5 ± 12.7 (≥18)	UAS7 ≥ 12 at week 32 in NCT02477332	240 mg Q4W	52 weeks	48 weeks	46% UAS7 = 0 at week 12 53% UAS7 = 0 at week 52	84% ≥ 1 AE 77% mild/moderate and mostly drug unrelated	Yes
NCT03437278 [40]	Phase 2 RDBPCT	49	12–17	UAS7 ≥ 16 HSS7 ≥ 8	24 mg or 120 mg Q4W, or 8 weeks placebo followed by 120 mg	24 weeks	16 weeks	↓ UAS7, HSS7, ISS7 UAS7 = 0 at week 24 (33% vs. 62% vs. 33%)	77% AEs 4% SAEs Nasopharyngitis Headache, Nausea	Yes
**Quilizumab**										
**Authors**	**Type of study**	**N.**	**Age (Yrs)**	**Indication**	**Dosage**	**Duration**	**Follow-Up**	**Results**	**Adverse events**	**Beneficial**
Harris et al. 2016 [41]	RDBPCT QUAIL study	32	18–75	CSU refractory to H1AH ± LTRAs UAS7 ≥ 16	450 mg or placebo Q4W	20 weeks	8 weeks	At week 20 ΔISS7 (−12.9, NS, *p* = 0.17) ΔUAS7 (−6, NS, *p* = 0.24)	No	No
**Mepolizumab**										
**Authors**	**Type of study**	**N.**	**Age (Yrs)**	**Indication**	**Dosage**	**Duration**	**Follow-Up**	**Results**	**Adverse events**	**Beneficial**
Magerl et al. 2018 [42]	Case report	1	27	Severe refractory eosinophilic asthma and refractory CSU	100 mg Q4W	16 weeks	NR	↑ UCT CSU remission Relapse after discontinuation	Discontinuation because of immune-complex reaction	Yes
**Reslizumab**										
**Authors**	**Type of study**	**N.**	**Age (Yrs)**	**Indication**	**Dosage**	**Duration**	**Follow-Up**	**Results**	**Adverse events**	**Beneficial**
Maurer et al. 2017 [43]	Case report	1	43	Severe refractory eosinophilic asthma and refractory CSU and cold urticaria	300 mg monthly	5 months	No	↑ UCT	NR	Yes
**Benralizumab**										
**Authors**	**Type of study**	**N.**	**Age (Yrs)**	**Indication**	**Dosage**	**Duration**	**Follow-Up**	**Results**	**Adverse events**	**Beneficial**
Bernstein A. et al. 2020 [44]	Single- blind trial	12	47.3 ± 1.3	CSU refractory to H1AH UAS ≥ 16	30 mg monthly after a dose of placebo	3 months	2 months	At week 20 ↓ UAS7 (−15.7) 3 (25%) withdrew (1 non-response)	No	Yes
**Dupilumab**										
**Authors**	**Type of study**	**N.**	**Age (Yrs)**	**Indication**	**Dosage**	**Duration**	**Follow-Up**	**Results**	**Adverse events**	**Beneficial**
Lee et al. 2019 [45]	Case series	6	36.2 (18–50)	CSU refractory to omalizumab up to 600 mg and H1AH (comorbidities: all AD, 1 asthma, 1 AH, 1 joint pain)	600 mg loading dose, then 300 mg Q2W Combined with H1AH	3 months	NR	Symptom resolution (3) ↓ UAS7 ≤ 6 (2) NR (1)	NR	Yes
Staubach et al. 2022 [46]	Case series	2	6–17	Inadequate response to H1AH, omalizumab (450 or 600 mg), and cyclosporine	300 mg Q2W	3 months	2–3 months	P1 UAS7 = 0 at week 8 P2 improvement at month 3	NR	Yes
Errichetti et al. 2021 [47]	Case series	2	52–63	CSU refractory to H1AH, LTRA, methotrexate, omalizumab, cyclosporine (comorbidities: Graves and atopic dermatitis)	600 mg, followed by 300 mg weekly	8 weeks	5–23 months	Complete response at week 8 and symptom free at follow-up	No	Yes
**Rituximab**										
**Authors**	**Type of study**	**N.**	**Age (Yrs)**	**Indication**	**Dosage**	**Duration**	**Follow-Up**	**Results**	**Adverse events**	**Beneficial**
Arkwright et al. 2009 [48]	Case report	1	12	CSU refractory to H1AH CSS dependence and side effects	375 mg/m^2^ weekly	4 doses	12 months	Symptom resolution for 12 months	NR	Yes
Chakravarty et al. 2011 [49]	Case report	1	51	CSU refractory to H1AH, H2AH, CSS, cyclosporine, mycophenolate mofetil	375 mg/m^2^ weekly Plus methotrexate	4 weeks	9 months	Symptom resolution for 8 months	NR	Yes
Steinweg et al. 2015 [50]	Case report	1	38	CSU refractory to H1AH and CSS	1000 mg QW2	2 weeks	10 months	Symptom resolution for 10 months	Fatigue Arthralgia Injection site-reaction	Yes
Combalia et al. 2018 [51]	Case report	1	44	Antisynthetase syndrome and CSU refractory to H1AH and immunosuppressants	1000 mg QW2 Plus one-week CSS	2 weeks	8 months	Early symptom resolution Mild controlled flares during follow-up	No	Yes
**Secukinumab**										
**Authors**	**Type of study**	**N.**	**Age (Yrs)**	**Indication**	**Dosage**	**Duration**	**Follow-Up**	**Results**	**Adverse events**	**Beneficial**
Sabag et al. 2020 [52]	Case series	8	NR	CSU refractory to H1AH, omalizumab, CSS, and cyclosporine UAS 32–40	150 mg weekly for 4 weeks then Q2W	3 months	NR	At day 30 ↓ 55% in UAS7 (−19.6) At day 90 ↓ 82% in UAS7 (−29.5)	Mild injection site reactions (3)	Yes
**Canakinumab**										
**Authors**	**Type of study**	**N.**	**Age (Yrs)**	**Indication**	**Dosage**	**Duration**	**Follow-Up**	**Results**	**Adverse events**	**Beneficial**
Maul et al. 2021 [53]	Phase 2 RDPCT	20	40.4 (18–70)	CSU refractory to H1AH ± CSS or LTRAs	150 mg	1 dose	8 weeks	Δ UAS7 (NS) Δ DLQI (NS)	No	No
**Infliximab**										
**Authors**	**Type of study**	**N.**	**Age (Yrs)**	**Indication**	**Dosage**	**Duration**	**Follow-Up**	**Results**	**Adverse events**	**Beneficial**
Wilson et al. 2011 [54]	Case report	1	35	CSU refractory to H1AH and immunosuppressants	5 mg/kg Q6W	NR	NR	Symptom free for 3 years, then flares controlled by cyclosporine	NR	Yes

AIS7, weekly activity interference score; CSS, corticosteroids; Δ, change from baseline; H2AH, H2 antihistamines; LTRAs, leukotriene receptor antagonists; N., number of patients; Q2W, every 2 weeks; Q4W, every 4 weeks; NR, not reported; NS, non-significant; SIS, sleep interference score; UCT, urticaria control test.

**Table 3 jcm-11-04453-t003:** List of the studies on biologics in CSU reporting disease severity score as an outcome.

Authors	Type of Study	N.	Dosage	End Point	Outcome
**Omalizumab**					
Saini et al. 2011 [25]	Phase 2 RDBPCT MYSTIQUE	90	75, 300, 600 mg combined with H1AH as needed	Week 4	↓ UAS7 (−9.8 vs. −19.9 vs. −14.6)
Maurer et al. 2011 [26]	RDBPCT X-QUISITE	49	75–375 mg Q2W or Q4W	Week 24	↓ UAS7 (−17.8)
Kaplan et al. 2013 [27]	Phase 3 RDBPCT GLACIAL	335	300 mg Q4W as add-on	Week 12	↓ ISS7 (−8.6) ↓ UAS7 (−19)
Maurer et al. 2013 [28]	RDBPCT ASTERIA II	323	75 mg, 150 mg, 300 mg Q4W	Week 12	↓ ISS7 (−5.9 vs. −8.1 vs. −9.8)
Saini et al. 2015 [29]	Phase 3 RDBPCT ASTERIA I	319	75 mg, 150 mg, or 300 mg Q4W	Week 12	↓ ISS7 (−6.4 vs. −6.6 vs. −9.4) ↓ UAS7 (−13.8 vs. −14.4 vs. −20.8)
Staubach et al. 2016 [30]	Phase 3 RDBPCT X-ACT	91	300 mg Q4W	Week 12	↓ UAS7 (−16)
Staubach et al. 2017 [31]	RDBPCT X-ACT	91	300 mg Q4W	Week 12	↓ AAS7 (−14.1)
Hide et al. 2017 [32]	Phase 3 RDBPCT POLARIS	218	150 mg, 300 mg Q4W With H1AH	Week 12	↓ ISS 7 (LSM) (−8.8 vs. −10.2) ↓ UAS 7 (LSM) (−18.8 vs. −22.4)
Yuan et al. 2022 [36]	RDBPCT	418	150 or 300 mg Q4W	Week 12	↓ ISS 7 (LSM) (−3.8 vs. −4.2)
**Ligelizumab**					
Maurer et al. 2019 [37]	Phase 2b RDBPCT NCT02477332	382	Ligelizumab 240 or 72 or 24 mg Q4W or omalizumab 300 mg Q4W; placebo Q4W; 120 mg ligelizumab followed by placebo Q4W	Week 12	HSS7 = 0 72 mg ligeliz > omaliz (51 vs. 26%) 240 mg ligeliz > omaliz (42 vs. 26%) UAS7 = 0 72 mg ligeliz > omaliz (44 vs. 26%) 240 mg ligeliz > omaliz (40 vs. 26%)
Maurer M et al. 2021 [39]	Open-label extension study of NCT02477332	226	240 mg Q4W	Week 12	UAS7 = 0 (41.6%)
NCT03437278 [40]	Phase 2 RDBPCT	49	24 mg or 120 mg Q4W, or 8 weeks placebo followed by 120 mg	Week 24	↓ UAS7 (−20.4 vs. −22.5 vs. −21.3)
**Quilizumab**					
Harris et al. 2016 [41]	RDBPCT QUAIL study	32	450 mg Q4W	Week 20	ISS7 (−12.9, NS)
**Mepolizumab**					
Magerl et al. 2018 [42]	Case report	1	100 mg Q4W	Week 12	↑ UCT
**Reslizumab**					
Maurer et al. 2017 [43]	Case report	1	300 mg monthly	Week 4	↑ UCT (+10)
**Benralizumab**					
Bernstein et al. 2020 [44]	Single- blind trial	12	30 mg monthly after a dose of placebo	Week 20	↓ UAS7 (−15.7)
**Dupilumab**					
Lee et al. 2019 [45]	Case series	6	600 mg loading dose, then 300 mg Q2W Combined with H1AH	Month 3 post-dupilumab	Symptom resolution (3) ↓ UAS7 ≤ 6 (2) NA (1)
Staubach et al. 2020 [46]	Case series	2	300 mg Q2W	NA	P1 UAS7 = 0 at week 8 P2 improvement at month 3
Errichetti et al. 2021 [47]	Case series	2	600 mg, followed by 300 mg weekly	NA	Complete response at week 8 and symptom free at follow-up
**Rituximab**					
Arkwright et al. 2009 [48]	Case report	1	375 mg/m^2^ Weekly	NA	Symptom resolution for 12 months
Chakravarty et al. 2011 [49]	Case report	1	375 mg/m^2^ weekly Plus mtx	NA	Symptom resolution for 8 months
Steinweg et al. 2015 [50]	Case report	1	1000 mg QW2	NA	Symptom resolution for 10 months
Combalia et al. 2018 [51]	Case report	1	1000 mg QW2 Plus one-week CSS	NA	Early symptom resolution Mild controlled flares during follow-up
**Secukinumab**					
Sabag et al. 2020 [52]	Case series	8	150 mg weekly for 4 weeks then Q2W	Day 90	↓ 82% in UAS7 (−29.5)
**Canakinumab**					
Maul et al. 2021 [53]	Phase 2 RDPCT	20	150 mg single dose	Week 4	Δ UAS7 (NS)
**Infliximab**					
Wilson et al. 2011 [54]	Case report	1	5 mg/kg Q6W	NA	Symptom free for 3 years, then flares controlled by cyclosporine

AAS7, weekly angioedema activity score; Δ, change from baseline; Q4W, every 4 weeks; LSM, least square means; N., number of patients; NA, not applicable; NS, not significant (*p* > 0.05).

**Table 4 jcm-11-04453-t004:** List of the studies on biologics in CSU reporting the HR-QoL score as an outcome.

Authors	Type of Study	N.	Dosage	End Point	Outcome
**Omalizumab**					
Maurer et al. 2011 [26]	RDBPCT X-QUISITE	49	75–375 mg Q2W or Q4W	Week 24	↓ DLQI ↓ Cu-Q2oL
Kaplan et al. 2013 [27]	Phase 3 RDBPCT GLACIAL	335	300 mg Q4W as add-on	Week 12	↓ DLQI (−9.7) ↓ CU-Q2OL (−29.3)
Maurer et al. 2013 [28]	RDBPCT ASTERIA II	323	75 mg, 150 mg, 300 mg Q4W + H1AH	Week 12	↓ DLQI (−7.5 vs. −8.3 vs. −10.2)
Saini et al. 2015 [29]	Phase 3 RDBPCT ASTERIA I	319	300 mg Q4W	Week 12	↓ DLQI (−10.3)
Staubach et al. 2016 [30]	Phase 3 RDBPCT X-ACT	91	300 mg or placebo Q4W	Week 28	↓ CU-Q2oL (LSM) (−21.5) ↓ DLQI (−10.5)
Staubach et al. 2017 [31]	RDBPCT X-ACT	91	300 mg or placebo Q4W	Week 4	↓ DLQI (LSM) (−7.6)
Hide et al. 2017 [32]	Phase 3 RDBPCT POLARIS	218	150 mg, 300 mg Q4W With H1AH	Week 12	↓ DLQI (−7.2 vs. −8.4)
Casale et al. 2019 [34]	Open-label + RDBPCT XTEND-CIU	205	300 mg Q4W for 24 weeks then randomization if UAS7 ≤ 6	Week 24	↓ DLQI (−12.6)
**Ligelizumab**					
Maurer et al. 2019 [37]	Phase 2b RDBPCT NCT02477332	382	Ligelizumab 240 or 72 or 24 mg Q4W or omalizumab 300 mg Q4W; placebo Q4W; 120 mg ligelizumab followed by placebo Q4W	Week 20	↓ DLQI (LSM) (−9.79 vs. −9.93 vs. −8.35 vs. −6.99)
Giménez-Arnau et al. 2022 [38]	Open-label extension study of NCT02477332	226	240 mg Q4W	Week 52	↓ DLQI (−9.52)
NCT03437278 [40]	Phase 2 RDBPCT	49	24 mg or 120 mg Q4W, or 8 weeks placebo followed by 120 mg	Week 12	↓ DLQI (−10.1 vs. −6.6 vs. −5)
**Canakinumab**					
Maul et al. 2021 [53]	Phase 2 RDPCT	20	150 mg single dose	Week 4	Δ DLQI (NS)

CSS, corticosteroids; Q4W, every 4 weeks; Δ, change from baseline; LSM, least square means; N., number of patients; NS, not significant (*p* > 0.05).

**Table 5 jcm-11-04453-t005:** List of the ongoing trials investigating mAbs in treating CSU.

**Ligelizumab**								
**Trial Number**	**Type of Study**	**Status**	**N.**	**Age (Yrs)**	**Inclusion Criteria**	**Dosage**	**Duration**	**Follow-Up**
NCT03907878 [55]	Phase 3 Multi-center, Open-label	Completed	66	≥18	CSU refractory to H1AH at approved doses UAS7 ≥ 16 and HSS7 ≥ 8	NR	52 weeks	12 weeks
NCT04210843 [56]	Phase 3 Double-blinded and open-label extension study Re-treatment with ligelizumab	Active, not recruiting	1041	≥12	CSU patients who successfully completed studies CQGE031C2302, CQGE031C2303, CQGE031C2202, or CQGE031C1301	72 mg followed by 120 mg Q4W or 120 mg Q4W	NR	NR
NCT04513548 [57]	Phase 1 RDBPCT (Part 2)	Active, not recruiting	68	18–79	CSU refractory to H1-AH UAS7 ≥ 16 and HSS7 ≥ 8 or cholinergic urticaria or cold urticaria	120 mg Q4W	16 weeks	12 weeks
NCT03580369 [58]	Phase 3 Multi-center RDBPCT	Active, not recruiting	1073	≥12	CSU refractory to H1-AH at approved doses UAS7 ≥ 16 and HSS7 ≥ 8	Ligelizumab Q4W or omalizumab 300 mg Q4W or placebo till week 20 followed by ligelizumab	52 weeks	12 weeks
NCT03580356 [59]	Phase 3 Multi-center RDBPCT	Active, not recruiting	1078	≥12	CSU refractory to approved doses of H1-AH UAS7 ≥ 16 and HSS7 ≥ 8	Ligelizumab Q4W or omalizumab 300 mg Q4W or placebo till week 20 followed by ligelizumab	52 weeks	12 weeks
**UB-221**								
**Trial number**	**Type of study**	**Status**	**N.**	**Age (Yrs)**	**Inclusion criteria**	**Dosage**	**Duration**	**Follow-Up**
NCT03632291 [60]	Phase 1 Open-label study	Completed	15	20–65	CSU	0.2 or 0.6 or 2 or 6 or 10 mg/kg	NR	NR
**Mepolizumab**								
**Trial number**	**Type of study**	**Status**	**N.**	**Age (Yrs)**	**Inclusion criteria**	**Dosage**	**Duration**	**Follow-Up**
NCT03494881 [61]	Phase 1 Open-label study	Recruiting	20	≥18	CSU refractory to H1AH	100 mg Q2W	8 weeks	NR
**Benralizumab**								
**Trial number**	**Type of study**	**Status**	**N.**	**Age (Yrs)**	**Inclusion criteria**	**Dosage**	**Duration**	**Follow-Up**
NCT04612725 [62]	Phase 2 RDBPCT	Active, not recruiting	155	≥18	CSU refractory to H1AH UAS7 ≥ 16 and ISS7 ≥ 8	NR	24 weeks with 28-week extension	NR
**Dupilumab**								
**Trial number**	**Type of study**	**Status**	**N.**	**Age (Yrs)**	**Inclusion criteria**	**Dosage**	**Duration**	**Follow-Up**
NCT03749135 [63]	Phase 2a RDBPCT	Completed	72	18–75	CSU refractory to standard trt UAS7 ≥ 16	NR	16 weeks	16 weeks
NCT04180488 [64]	Phase 3 Multi-center RDBPCT	Active, not recruiting	384	6–80	CSU refractory to H1AH UAS7 ≥ 16 and ISS7 ≥ 8 Study A: omalizumab naïve Study B: intolerant or incomplete responder to omalizumab	NR	24 weeks	12 weeks
**Rituximab**								
**Trial number**	**Type of study**	**Status**	**N.**	**Age (Yrs)**	**Inclusion criteria**	**Dosage**	**Duration**	**Follow-Up**
NCT00216762 [65]	Phase 1–2 Open-label	Terminated	15	18–70	CSU refractory to high dose H1AH and immuno suppressants	1000 mg Q2W	2 weeks	NR
**Tezepelumab**								
**Trial number**	**Type of study**	**Status**	**N.**	**Age (Yrs)**	**Inclusion criteria**	**Dosage**	**Duration**	**Follow-Up**
NCT04833855 [66]	Phase 2b RDBPCT	Recruiting	159	18–80	CSU refractory to H1AH and 6-months omalizumab UAS7 ≥ 16 and HSS7 ≥ 8	NR	16 weeks	NR
**Barzolvolimab**								
**Trial number**	**Type of study**	**Status**	**N.**	**Age (Yrs)**	**Inclusion criteria**	**Dosage**	**Duration**	**Follow-Up**
NCT04538794 [67]	Phase 1 RDBPCT	Recruiting	159	18–75	CSU refractory to H1AH ± H2AH or LTRAs UAS7 ≥ 16 and ISS7 ≥ 8	NR	12 weeks	12 weeks
NCT05368285 [68]	Phase 2 RDBPCT	Recruiting	168	≥18	CSU refractory to H1AH UAS7 ≥ 16 and ISS7 ≥ 8	A. 75 mg for 16 weeks then 150 mg Q4W B. 75 mg for 16 weeks then 300 mg Q4W C. 150 mg Q4W D. 300 mg Q8W E. 16-weeks placebo then 150 mg Q4W F. 16-weeks placebo then 300 mg Q4W	52 weeks	NR
**MTPS9579A**								
**Trial number**	**Type of study**	**Status**	**N.**	**Age (Yrs)**	**Inclusion criteria**	**Dosage**	**Duration**	**Follow-Up**
NCT05129423 [69]	Phase 2 Multi-center RDBPCT	Recruiting	240	18–75	CSU refractory to H1AH	Part 1 (12 weeks): Dose A vs. placebo Q4W Part 2 (12 weeks): Dose A, B, C, D vs. placebo Q4W	24 weeks	NR
**LY3454738**								
**Trial number**	**Type of study**	**Status**	**N.**	**Age (Yrs)**	**Inclusion criteria**	**Dosage**	**Duration**	**Follow-Up**
NCT04159701 [70]	Phase 2 RDBPCT	Terminated for lack of efficacy	52	18–65	CSU refractory to H1AH	A 500 mg Q2W for 12 weeks followed by placebo B Placebo for 12 weeks followed by 500 mg Q2W	24 weeks	NR
**Lirentelimab**								
**Trial number**	**Type of study**	**Status**	**N.**	**Age (Yrs)**	**Inclusion criteria**	**Dosage**	**Duration**	**Follow-Up**
NCT03436797 [71]	Phase 2 Open-label study	Completed	47	18–85	CU refractory to H1AH	Up to 3 mg/kg Q4W	6 months	8 weeks

**Table 6 jcm-11-04453-t006:** Small molecules inhibitors and their target.

** *Biological Drugs* **	** *Target* **
** *Bruton’s tyrosine kinase (BTK) inhibitors* **	
**Remibrutinib (LOU064)**	BTK
**Rilzabrutinib**	BTK
**Tirabrutinib**	BTK
**Fenebrutinib (GDC-0853)**	BTK
**TAS5315**	BTK
** *Others* **	
**Etanercept**	TNF-α
**TLL018**	JAK1/TYK2
**AZD1981**	Prostaglandin D2 receptor 2 (DP2 or CRTH2)

**Table 7 jcm-11-04453-t007:** List of the trials on small molecules inhibitors.

Trial Number	Type of Study	Status	N.	Age (Yrs)	Inclusion Criteria	Duration
**Remibrutinib**						
NCT03926611 [134]	Phase 2 RDBPCT	Completed	311	≥18	CSU refractory to H1AH UAS ≥ 16	12 weeks
NCT04109313 [135]	Open label	Active, not recruiting	195	18–99	Completed CLOU064A2201 or other preceding studies with LOU064	52 weeks
NCT05048342 [136]	Phase 3 Open label	Recruiting	70	≥18	CSU refractory to H1AH UAS ≥ 16	52 weeks
NCT05032157 NCT05030311 [137,138]	Phase 3 RDBPCT	Recruiting	450	≥18	CSU refractory to H1AH UAS ≥ 16	24 weeks +28 weeks
NCT05170724 [139]	Cohort	Available	NR	18–99	CSU refractory to H1AH	NR
**Fenebrutinib (GDC-0853)**						
Metz et al. 2021 [140]	Phase 2 RDBPCT	Completed	134	18–75	CSU refractory to H1AH	8 weeks
NCT03693625 [141]	Phase 2 Open label	Terminated	31	18–75	CSU refractory to H1AH	NR
**Tirabrutini** **b**						
NCT04827589 [142]	Phase 2 RDBPCT	Withdrawn	NR	18–75	CSU refractory to H1AH UAS ≥ 16	8 weeks +16 weeks
**Rilzabrutinib**						
NCT05107115 [143]	Phase 2 RDBPCT + OL	Recruiting	152	18–80	CSU refractory to H1AH	12 weeks +40 weeks
**TAS5315**						
NCT05335499 [144]	Phase 2 RDBPCT	Not yet recruiting	120	18–75	CSU refractory to H1AH UAS ≥ 16	12 weeks
**Etanercept**						
NCT01030120 [145]	RDBPCT Open label	Withdrawn	0	18–70	CSU refractory to H1AH	6 weeks +6 weeks
**TLL018**						
NCT05373355 [146]	Phase 1 RDBPCT	Not yet recruiting	36	18–70	CSU and UAS ≥ 16	12 weeks
**AZD1981**						
Oliver et al. 2019 [147]	Phase 2 RDBPCT	Completed	26	18–65	CSU refractory to H1AH	8 weeks

## Data Availability

Not applicable.

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
