# Peer review of "Monoclonal Antibodies in Treating Chronic Spontaneous Urticaria: New Drugs for an Old Disease"

_jcm, 2022, doi:10.3390/jcm11154453_

Round 1

Reviewer 1 Report

Thanks to the authors for summarizing the monoclonal antibodies options to treat Chronic Spontaneous Urticaria. I think this review paper has potential to be the interest of many readers as long as it presented differently.

1)It needs a figure or diagram indicating where each medication acts in the pathophysiology of CSU. This has been done for many biologics in other diseases (e.g. psoriasis)

2)Table comparing all biologic using dosage and primary outcome

3)Table comparing secondary outcomes (for example pruritus or QOL)

Based on the available data which biologic seems to be the more promising. An input from teh authors based on their experience would be also helpful. How do you foresee the treatment of CSU? Combination therapy would be the gold standard? (antihistamines + biologics?)

Author Response

Dear Editor,  

Thank you for your review of our manuscript entitled “Monoclonal Antibodies in treating Chronic Spontaneous Urticaria: new drugs for an old disease” (Manuscript ID: jcm-1813249). We greatly appreciate the constructive comments. We have read your comments and those of the reviewers and have revised the manuscript. Below, I would like to outline our responses to these comments.

We used point-to-point response to reviewers’ comments. In addition, you will find, colored in red, suggested modifications in the text.

Thank you again for your interest in our work. We hope that this revision meets with your approval. We await your review of our revised manuscript.

Reviewer 1: Thanks to the authors for summarizing the monoclonal antibodies options to treat Chronic Spontaneous Urticaria. I think this review paper has potential to be the interest of many readers as long as it presented differently.
1) It needs a figure or diagram indicating where each medication acts in the pathophysiology of CSU. This has been done for many biologics in other diseases (e.g. psoriasis)
Response: We replaced table 4 with a figure, showing targets of mAbs.

2) Table comparing all biologic using dosage and primary outcome

Response: Thank you. We added a table including studies on biologics that reported a disease severity as outcome.

3) Table comparing secondary outcomes (for example pruritus or QOL)

Response: As requested, we added a table including studies on biologics that reported data about QoL as assessed outcome

Based on the available data which biologic seems to be the more promising. An input from teh authors based on their experience would be also helpful. How do you foresee the treatment of CSU? Combination therapy would be the gold standard? (antihistamines + biologics?)

Response: Omalizumab is widely recognized as an effective treatment for refractory CSU. In the future, treatment of CSU will be individualized based on the underlying pathogenic mechanism. In this context, mAbs targeting upstream the inflammatory cascade of CSU might be an alternative treatment, especially for CSU refractory to omalizumab. The role of H1AH may be that of rescue medication in the event of relapses of CSU.

Reviewer 2 Report

This is a review manuscript intended to describe the current status of in use or under investigation of mabs in CSU. Other new drugs, like kinase inhibitors, were excluded.

But some points need to be addressed:

Abstract: Results section not adequate and modifications are advised. Not usual to include discussion in abstract.

Introduction line 65: “However, in approximately half of the patients suffering from CU, the causing factor can not be detected”. it is estimated that about 70% of CU is spontaneous and 30% inducible. (reference one did not cite this percentage).

Introduction line 90: “The treatment of CSU is based on symptomatic drugs and second-generation H1-antihistamines (H1AH)”. Which symptomatic drugs are those?

Results:

Tables (1, 2, 3): None of the studies were adequately referred, so it was almost impossible to find them. It is of great importance to refer each study, enabling readers to check the correct inclusion of the study.

One example of this issue is that at Table 1/dupilumab: study by Lee et al not included in References. And I didn’t find the study by Staubach et al, not even one on pubmed/medline by searching Staubach & dupilumab.

The authors are advised to check if all papers were included in reference. Additionally, and based on that, it may be necessary to review the results.

Table 2/status: authors are advised to update status as some of these trials cited as "recruiting" are currently "active, not recruiting", especially for ligelizumab.

Line 192: Any difference… is that “no difference”?

Line 313: any serious adverse… is that “no serious adverse”?

Line 323: Nevertheless, any significant change was… is that “no significant change”?

Line 380: A trial only is available about benralizumab. is that “One trial only is available…”?

Line 193: weekly Itchy severity scale (ISS7). Correct is weekly Itch Severity Score (ISS7).

Line 206: “The efficacy of omalizumab at week 12 in ASTERIA I (n=319) was comparable with the previous trials as regards ISS7, UAS7 and DLQI.” The word “previous” can be confusing, as, at least GLACIAL trail, was published later, maybe changing to "the above mentioned" (or anything similar) would be better.

Line 222: “…has been few investigated so far.” not used. suggestion: has been less investigated so far.

3.1.2. Ligelizumab: The authors claim that ligelizumab is superior to omalizumab in CSU, and indeed early studies suggested it. But recently, Novartis declared that studies showed superiority of ligelizumab vs placebo in CSU, but not to omalizumab. Apparently, ligelizumab won't be marketed for CSU. (https://www.novartis.com/news/media-releases/novartis-provides-update-phase-iii-ligelizumab-qge031-studies-chronic-spontaneous-urticaria-csu)

3.3.1. Dupilumab: EMA for individuals with moderate-to-severe uncontrolled atopic dermatitis (12 years old or, if severe, 6 years old), severe uncontrolled asthma (6 years old).

FDA approved for 6 months for AD.

Lines 434/435. Strange sentence, please check

Discussion:

Table 4:

Ligelizumab: IgE - high-affinity IgE receptors (FcεRIα). confusing: High-affinity anti-IgE.

Long-acting anti-IgEs. Why IgEs and not IgE?

Line 530: EAACI guidelines [1, 116]. is the reference 116 correct? it´s not EAACI guideline, nor about omalizumab.

Authors must review all reference on the text.

Author Response

Dear Editor,  

Thank you for your review of our manuscript entitled “Monoclonal Antibodies in treating Chronic Spontaneous Urticaria: new drugs for an old disease” (Manuscript ID: jcm-1813249). We greatly appreciate the constructive comments. We have read your comments and those of the reviewers and have revised the manuscript. Below, I would like to outline our responses to these comments.

We used point-to-point response to reviewers’ comments. In addition, you will find, colored in red, suggested modifications in the text.

Thank you again for your interest in our work. We hope that this revision meets with your approval. We await your review of our revised manuscript.

Reviewer 2: This is a review manuscript intended to describe the current status of in use or under investigation of mabs in CSU. Other new drugs, like kinase inhibitors, were excluded. But some points need to be addressed:

Abstract: Results section not adequate and modifications are advised. Not usual to include discussion in abstract.

Response: Thank you. We reviewed the results section and removed the discussion section in the abstract.

Introduction line 65: “However, in approximately half of the patients suffering from CU, the causing factor can not be detected”. it is estimated that about 70% of CU is spontaneous and 30% inducible (reference one did not cite this percentage).

Response: Thank you, we provided an adequate reference on the incidence of CSU.

Introduction line 90: “The treatment of CSU is based on symptomatic drugs and second-generation H1-antihistamines (H1AH)”. Which symptomatic drugs are those?

Response: H1-antihistamines are symptomatic drugs, accordingly, we modified the sentence.

Results: Tables (1, 2, 3): None of the studies were adequately referred, so it was almost impossible to find them. It is of great importance to refer each study, enabling readers to check the correct inclusion of the study.

Response: Thank you. We added references in each study/trial reported in the tables, as suggested.

One example of this issue is that at Table 1/dupilumab: study by Lee et al not included in References. And I didn’t find the study by Staubach et al, not even one on pubmed/medline by searching Staubach & dupilumab.

Response: The study by Lee had been already included in References (please, see reference 45).

The study by Staubach can be found by searching dupilumab AND chronic spontaneous urticaria on Pubmed database: Staubach P, Peveling-Oberhag A, Lang BM, Zimmer S, Sohn A, Mann C. Severe chronic spontaneous urticaria in children - treatment options according to the guidelines and beyond - a 10 years review. J Dermatolog Treat. 2022;33(2):1119-1122. doi:10.1080/09546634.2020.1782326

The authors are advised to check if all papers were included in reference. Additionally, and based on that, it may be necessary to review the results.

Response: Thank you, we checked and reviewed all the references included in the manuscript.

Table 2/status: authors are advised to update status as some of these trials cited as "recruiting" are currently "active, not recruiting", especially for ligelizumab.

Response: Thank you. We checked the included trials on clin.gov and we updated their status, since some of them received an update after June 30, 2022.

Line 192: Any difference… is that “no difference”?

Line 313: any serious adverse… is that “no serious adverse”?

Line 323: Nevertheless, any significant change was… is that “no significant change”?

Line 380: A trial only is available about benralizumab. is that “One trial only is available…”?

Line 193: weekly Itchy severity scale (ISS7). Correct is weekly Itch Severity Score (ISS7).

Line 206: “The efficacy of omalizumab at week 12 in ASTERIA I (n=319) was comparable with the previous trials as regards ISS7, UAS7 and DLQI.” The word “previous” can be confusing, as, at least GLACIAL trail, was published later, maybe changing to "the above mentioned" (or anything similar) would be better.

Line 222: “…has been few investigated so far.” not used. suggestion: has been less investigated so far.

Response: Thank you. We corrected, as suggested.

3.1.2. Ligelizumab: The authors claim that ligelizumab is superior to omalizumab in CSU, and indeed early studies suggested it. But recently, Novartis declared that studies showed superiority of ligelizumab vs placebo in CSU, but not to omalizumab. Apparently, ligelizumab won't be marketed for CSU. (https://www.novartis.com/news/media-releases/novartis-provides-update-phase-iii-ligelizumab-qge031-studies-chronic-spontaneous-urticaria-csu)

Response: Thank you for the suggestion. This update by Novartis was added in the main text (discussion), though data have not been released yet.

3.3.1. Dupilumab: EMA for individuals with moderate-to-severe uncontrolled atopic dermatitis (≥12 years old or, if severe, ≥6 years old), severe uncontrolled asthma (≥6 years old). FDA approved for ≥6 months for AD.

Response: Thank you. We updated the new indications for dupilumab recently approved by the FDA.

Lines 434/435. Strange sentence, please check

Response: Thank you. We modified the sentence, as suggested

Discussion:

Table 4:

Ligelizumab: IgE - high-affinity IgE receptors (FcεRIα). confusing: High-affinity anti-IgE.

Response: Thank you. Table 4 was removed and replaced with figure 1.

Long-acting anti-IgEs. Why IgEs and not IgE?

Thank you. We replaced anti-IgEs with anti-IgE, as suggested.

Line 530: EAACI guidelines [1, 116] is the reference 116 correct? it´s not EAACI guideline, nor about omalizumab.

Response: Thank you. Reference 116 was erroneously reported in that sentence.

Authors must review all reference on the text.

Response: Thank you, we checked and reviewed all the references included in the manuscript.

We are grateful to you and reviewers for the truly helpful comments.

These changes will improve the quality of our paper.

Sincerely,

Sara Manti, MD, Pediatrician

PhD, University Researcher

Pediatric Respiratory Unit, Department of Clinical and Experimental Medicine, San Marco Hospital, University of Catania, Via Santa Sofia 78, 95123, Catania, Italy

Pediatric Unit, Department of Human and Pediatric Pathology “Gaetano Barresi”, AOUP G. Martino, University of Messina, Via Consolare Valeria, 1, 98124 , Messina, Italy

E-mail: saramanti@hotmail.it

Mobile phone: 00393296924334